# Inactivation of the High-Affinity Ca^2+^ Uptake System Delays the Amiodarone-Induced Ca^2+^ Influx in Yeast *Ogataea parapolymorpha*

**DOI:** 10.3390/ijms262311386

**Published:** 2025-11-25

**Authors:** Maria Kulakova, Maria Pakhomova, Victoria Bidiuk, Michael Agaphonov

**Affiliations:** The Federal Research Center “Fundamentals of Biotechnology” of the Russian Academy of Sciences, 119071 Moscow, Russia

**Keywords:** high-affinity calcium uptake system, voltage-gated calcium channel, amiodarone, calcium homeostasis, yeast, *Ogataea*

## Abstract

The antiarrhythmic drug amiodarone is toxic to yeast cells due to provoking Ca^2+^ entry into cytosol. Here we show that in *Ogataea parapolymorpha*, the loss of Cch1 or Mid1, which are the primary components of the high-affinity Ca^2+^ uptake system (HACS), leads to a delay in the rise of cytosolic Ca^2+^ concentration ([Ca^2+^]_cyt_) in response to amiodarone. This has negligible effect on the ability of the strain with the unaffected Ca^2+^ sequestration system to grow in the presence of amiodarone. Inactivation of the *PMC1* gene encoding the Ca^2+^ ATPase involved in the cytosolic Ca^2+^ sequestration in the vacuole dramatically increases sensitivity to amiodarone, while inactivation of *CCH1* or *MID1* suppresses it. This correlates with a substantially lower [Ca^2+^]_cyt_ rise in response to amiodarone when the genes encoding the HACS components are inactivated in the mutant lacking Pmc1. Similarly to sodium dodecyl sulfate, which has also been shown to increase [Ca^2+^]_cyt_, amiodarone causes activation of the Hog1 protein kinase involved in the cell cycle regulation. The role of HACS in the amiodarone-induced Ca^2+^ influx is discussed.

## 1. Introduction

In the eukaryotic cell Ca^2+^ cations fulfill a number of functions that require their sequestration in certain organelles while their concentration in cytosol ([Ca^2+^]_cyt_) is maintained at a very low level. This is achieved due to the action of different ion pumps and channels. In yeast, the main Ca^2+^ sink/storage organelle is the vacuole, whose Ca^2+^ ATPase Pmc1 and H^+^/Ca^2+^ antiporter Vcx1 pump Ca^2+^ from the cytosol. Vcx1 is powered by the proton gradient across the vacuolar membrane created by the vacuolar H^+^ ATPase [1]. Unlike animal cells, yeast cells do not possess a special Ca^2+^ ion pump in the endoplasmic reticulum (ER), which receives these cations from the Golgi apparatus [2] and possibly from the vacuole [3]. The Golgi apparatus is supplied with Ca^2+^ by the Pmr1 Ca^2+^/Mn^2+^ ATPase, which is also involved in the control of [Ca^2+^]_cyt_ [4,5].

Two Ca^2+^ uptake systems, namely, low-affinity (LACS) and high-affinity (HACS), have been detected in yeast cells [6]. No components of LACS have been identified so far except Fig1 [7], whose exact function remains unknown. HACS is represented by Cch1, which is a homolog of the mammalian voltage-gated Ca^2+^ channels [8] and by Mid1, which shares some homology with the family of animal proteins associated with the sodium leak channels [9]. Besides these two proteins, Ecm7 has also been shown to be involved in the functioning of HACS [10].

The antiarrhythmic drug amiodarone is toxic to a wide range of fungi [11]. In *S. cerevisiae*, it was shown to provoke Ca^2+^ influx, which causes its toxicity [12,13,14]. The cell death induced by amiodarone in yeast demonstrates hallmarks similar to those of the cell death induced by the alpha-factor pheromone [15]. Notably, Cch1 and Mid1 are involved in the [Ca^2+^]_cyt_ rise in response to the alpha factor [8]. It was reported that in contrast to the loss of Cch1, the loss of Mid1 significantly reduced the [Ca^2+^]_cyt_ rise in response to amiodarone [13], although the other manifestations of these mutations are identical [16]. In humans, the action of amiodarone as a class III antiarrhythmic drug is achieved through inhibition of potassium channels, and it also blocks cardiac Na^+^ and Ca^2+^ channels (for a review see [17]).

In *Ogataea* yeasts, the loss of the vacuolar Ca^2+^ ATPase Pmc1 leads to hypersensitivity to sodium dodecyl sulfate (SDS) since the latter induces Ca^2+^ influx from the environment [18,19]. Although the *Saccharomyces cerevisiae pmc1-*Δ mutant is not hypersensitive to SDS [18], this phenotype in some mutants can also be related to Ca^2+^ homeostasis and signaling. Particularly, SDS sensitivity of the *vps13-*Δ mutant can be suppressed by inhibition of calcineurin, which is the Ca^2+^/calmodulin-dependent protein phosphatase regulating Ca^2+^ homeostasis and adaptation to environmental changes [20,21], or by an increased *PMC1* gene dosage [22,23]. The loss of Cch1 suppresses SDS hypersensitivity of the *pmc1-*Δ mutants but does not prevent SDS-induced Ca^2+^ uptake [18,19]. This indicates that the main conduit for Ca^2+^ entry in response to SDS is not the Cch1/Mid1 channel, but its regulation depends on Cch1. At the same time, the role of Mid1 in SDS-induced Ca^2+^ uptake has not been explored.

The objective of this work was to reveal the role of the Cch1 and Mid1 components of yeast HACS in Ca^2+^ entry in response to amiodarone using *Ogataea parapolymorpha* expressing a genetically encoded fluorescent Ca^2+^ indicator as a model. We have shown that the effects of *MID1* and *CCH1* inactivation on the [Ca^2+^]_cyt_ rise in response to amiodarone, as well as to SDS, are very similar and that the loss of the components of HACS inhibits only the initial stage of Ca^2+^ uptake induced by amiodarone.

## 2. Results

### 2.1. Inactivation of MID1 Suppresses SDS Hypersensitivity Caused by the Lack of the Pmc1 Ca^2+^ Ion Pump

While the Cch1 subunit of the high-affinity plasma membrane Ca^2+^ channel in *Ogataea* yeasts has been characterized previously [18,19], the Mid1 subunit remained uncharacterized. The only homolog of *S. cerevisiae MID1* in the *O. parapolymorpha* genome (NCBI gene ID: 25772629) is also annotated as *MID1* [24]. Interestingly, this ORF starts only 22 bp downstream of the ORF encoding alanyl-tRNA synthetase. This distance apparently is not sufficient to include the full promoter, which possibly overlaps with the upstream ORF. As in *S. cerevisiae* Mid1, the *O. polymorpha* polypeptide encoded by this gene starts with a hydrophobic region (Table 1, Appendix A), whose deletion in *S. cerevisiae* was shown to affect translocation of this protein to the endoplasmic reticulum [25]. Unlike the *S. cerevisiae* protein, the *O. parapolymorpha* Mid1 possesses a hydrophobic region at the C-terminus, which can potentially serve as a transmembrane domain. Notably, the Mid1 homologs of some other fungi, e.g., *Yarrowia lipolytica*, *Schizosaccharomyces pombe*, *Candida albicans* and *Aspergillus fumigatus* (Table 1), also possess the C-terminal hydrophobic region. Similar to the *S. cerevisiae* Mid1, the *Cryptococcus neoformans* protein contains only one hydrophobic region, which, however, does not start from the N-terminus but from the 58th amino acid residue. The *A. fumigatus* Mid1 contains both hydrophobic regions, and the N-terminal one starts from the 14th amino acid residue (Table 1, Appendix A). Although the N-terminal hydrophobic region was predicted to be the secretory signal peptide [25], its position in the *C. neoformans* and *A. fumigatus* proteins contradicts the role as a secretory signal. Moreover, this protein should be membrane-associated, which implies that it should possess a hydrophobic region, which is not cleaved off during maturation. That is why we suppose that this region is not a genuine secretory signal.

Previously we have shown that the loss of Cch1 suppresses SDS hypersensitivity of *O. polymorpha* and *O. parapolymorpha* mutants defective in the vacuolar Ca^2+^ ATPase Pmc1 [18,26]. To determine whether the loss of Mid1 can cause the same effect, the *MID1* gene was inactivated in the *O. parapolymorpha pmc1-*Δ strain. Indeed, the *pmc1-*Δ *mid1-*Δ double mutant was able to grow at higher SDS concentrations than the original *pmc1-*Δ strain. Notably, as with inactivation of *CCH1*, inactivation of *MID1* suppressed SDS sensitivity not only in the *pmc1-*Δ mutant but also in the strain with wild-type *PMC1* and caused the inability to grow on the Ca^2+^-deficient medium supplemented with ethylene glycol-bis(β-aminoethyl ether)-N,N,N′,N′-tetraacetate (EGTA) as a Ca^2+^ chelator (Figure 1a,b).

### 2.2. Effects of Inactivation of MID1 and CCH1 on the SDS-Induced [Ca^2+^]_cyt_ Rise in the pmc1-Δ Mutant Are the Same

Since we did not observe a noticeable difference between the ability of deletions of *MID1* and *CCH1* to suppress SDS sensitivity of the *pmc1-*Δ mutant, we reasoned that the effects of these mutations on the [Ca^2+^]_cyt_ rise in response to SDS in the *pmc1-*Δ would also be the same. To study this, we used the genetically encoded Ca^2+^ indicator GEM-GECO [27], which enables monitoring [Ca^2+^]_cyt_ in individual yeast cells using flow cytometry by measuring the fluorescence ratio at 450 nm and 525 nm (FL_450_/FL_525_) [26]. Indeed, we observed very similar dynamics of the [Ca^2+^]_cyt_ rise in response to SDS in the *pmc1-*Δ *cch1-*Δ and *pmc1-*Δ *mid1-*Δ mutants, which was less steep than that in the *pmc1-*Δ single mutant (Figure 2).

### 2.3. Amiodarone Causes a Rapid Increase in [Ca^2+^]_cyt_ and Hog1 Activation in O. parapolymorpha

According to the previous studies performed in *S. cerevisiae* using aequorin to monitor [Ca^2+^]_cyt_ [13,14], amiodarone causes an increase in [Ca^2+^]_cyt_. Here, we studied the effects of this drug on [Ca^2+^]_cyt_ in *O. parapolymorpha* using the GEM-GECO Ca^2+^ indicator. Similar to that in *S. cerevisiae*, amiodarone induced a [Ca^2+^]_cyt_ rise in *O. parapolymorpha*. This was essentially abolished by the presence of EGTA or ethylenediaminetetraacetate (EDTA) as a chelator in culture medium (Figure 3), indicating that Ca^2+^ is transported from the environment.

Similarly to that observed for the SDS treatment [19], the [Ca^2+^]_cyt_ rise in response to amiodarone treatment was accompanied by activation of the Hog1 protein kinase. Notably, the effect of amiodarone lasted longer than that of SDS but still was not as intensive and prolonged as in case of Hog1 activation in response to high osmolality of NaCl (Figure 4). This indicates that, similar to sensitivity to SDS [18], sensitivity to amiodarone can be mediated by the Hog1 protein kinase.

### 2.4. Effects of the Loss of HACS in the Strain with Wild-Type PMC1 and in the pmc1-Δ Mutant on Survival Rate on Amiodarone-Containing Medium Are Opposite

It was reasonable to expect that, similar to that observed in *S. cerevisiae* [14], the amiodarone-induced [Ca^2+^]_cyt_ rise in *O. parapolymorpha* cells may affect growth rate and cause cell death. Indeed, when dilutions of cell suspensions were spotted onto YPD supplemented with amiodarone concentrations ranging from 0.1 to 1 mM, some growth attenuation was observed at the 0.1 mM concentration of this drug, but the number of colonies on the spots was similar to that on the control plate (Appendix A). At higher amiodarone concentrations, a successive decrease in growth rate as well as some reduction in the number of colonies was observed; however, colonies emerged even with 1 mM amiodarone (Appendix A). This means that only some fractions of cells were unable to form colonies on the amiodarone-containing medium. To determine whether the proportion of cells able to survive in the presence of amiodarone depends on the HACS, exponentially growing cultures of the *mid1-*Δ and *cch1-*Δ mutants, as well as the wild-type control strain, were spread on YPD plated with and without amiodarone to count the numbers of colony-forming units as described in Materials and Methods. Inactivation of *MID1* or *CCH1* decreased the survival rate (Figure 5a); however, the attenuation of the colony growth on the medium supplemented with amiodarone was very similar in the mutants and wild-type control strain (Appendix A).

Notably, YPD supplemented with more than 0.5 mM amiodarone becomes visibly turbid, indicating a colloid state of amiodarone, which tended to precipitate onto the well surface after a while if the experiment was performed in a 96-well plate in liquid medium. That was why we could not reach the amiodarone concentration, which prevented growth of cells with the wild-type Ca^2+^ transport system since those cells were able to grow in liquid YPD containing more than 0.5 mM amiodarone. The lack of the Pmc1 vacuolar Ca^2+^ ATPase significantly decreased resistance to amiodarone, while the loss of Cch1 or Mid1 suppressed this amiodarone hypersensitivity (Figure 1c and Figure 5b).

### 2.5. Caffeine Does Not Alleviate Effects of Amiodarone on Cell Growth and Has a Negligible Effect on the Amiodarone-Induced [Ca^2+^]_cyt_ Rise

It was observed in a previous study [13], which employed aequorin to monitor Ca^2+^ concentration in *S. cerevisiae*, that the [Ca^2+^]_cyt_ rise in response to amiodarone is strongly inhibited in the presence of caffeine. In contrast to those data, the use of the GEM-GECO Ca^2+^ indicator revealed only a slight decrease in the FL_450_/FL_525_ ratio when cells were treated with amiodarone in the presence of caffeine (Figure 6), which could be due to some unrelated effect of caffeine on cell physiology. It was reasonable to expect that caffeine should rescue cells from amiodarone action if it inhibits the amiodarone-induced Ca^2+^ uptake. However, the presence of caffeine did not improve the survival or colony growth rate on amiodarone-containing medium in either *O. polymorpha* or *S. cerevisiae* (Figure 5a, Appendix A). This indicates that the amiodarone action is not inhibited by caffeine.

### 2.6. Deletion Mutations of MID1 and CCH1 Delay the Amiodarone-Induced Ca^2+^ Influx

To explore whether HACS is involved in the amiodarone-induced Ca^2+^ entry, the FL_450_/FL_525_ ratio was determined in strains expressing the GEM-GECO indicator. Notably, the rapid [Ca^2+^]_cyt_ rise in response to amiodarone was not immediate. In the strain with the wild-type Ca^2+^ metabolism, it was observed after 1–2 min of incubation with amiodarone and continued for more than 10 min. Inactivation of *MID1* or *CCH1* led to an additional delay in the amiodarone-induced [Ca^2+^]_cyt_ rise lasting approximately 3–5 min, but then [Ca^2+^]_cyt_ started rising, reaching similar levels to the wild-type strain (Figure 7a).

In the strain lacking the Pmc1 vacuolar Ca^2+^ ATPase, the initial [Ca^2+^]_cyt_ rise was not as steep as in the wild-type strain, but then it sped up and after prolonged incubation reached higher levels than in the wild-type strain (Figure 7b), apparently due to the inability to efficiently sequester cytosolic Ca^2+^ in the vacuole. Inactivation of either *MID1* or *CCH1* in the *pmc1-*Δ mutant increased the delay in the [Ca^2+^]_cyt_ rise, and [Ca^2+^]_cyt_ reached a noticeably lower level after prolonged incubation (Figure 7b). This is in agreement with the observation that the loss of HACS components suppresses *pmc1-*Δ hypersensitivity to amiodarone (Figure 1c and Figure 5b).

## 3. Discussion

As mentioned in the Introduction, the yeast plasma membrane high-affinity Ca^2+^ ion channel consists of two subunits, namely, Cch1 and Mid1. While the Cch1 predicted topology [28] and its homology to mammalian voltage-gated Ca^2+^ channels [8] indicate that it is the channel-forming subunit, the exact role of Mid1 in the complex remains unclear. Some published data indicate that the Mid1 subunit of the Cch1/Mid1 Ca^2+^ channel may have a function outside of this complex. Particularly, production of *S. cerevisiae* Mid1 alone in animal cells conferred sensitivity to mechanical stress [29]. The lack of Mid1, but not Cch1, affected the [Ca^2+^]_cyt_ rise in response to amiodarone in *S. cerevisiae*, while cell growth in the presence of amiodarone was improved by the loss of Cch1, but not Mid1 [13]. The effects of amiodarone on [Ca^2+^]_cyt_ in *S. cerevisiae* were studied using the Ca^2+^-dependent luciferase aequorin [13,14]. However, this approach does not allow [Ca^2+^]_cyt_ to be monitored in single cells. Previously [26], we have shown that the GEM-GECO Ca^2+^ indicator [27] can be used to monitor [Ca^2+^]_cyt_ in single cells of the *O. parapolymorpha* yeast, and this was applied here to study the role of these HACS components in reaction of yeast cells to amiodarone. Similar to previous observations made in *S. cerevisiae*, we have observed a substantial [Ca^2+^]_cyt_ rise in response to amiodarone in *O. parapolymorpha*; however, in contrast to that observed in *S. cerevisiae* [13], it did not depend on caffeine. There is strong evidence that the direct target of caffeine in yeast is Tor1 protein kinase [30,31], and how this compound can affect the Ca^2+^ transport system is unclear. We do not think that HACS reacts differently to caffeine in these two yeast species. Indeed, caffeine had the same effect on cell growth in both yeast species in our experiments. Caffeine interfering with Ca^2+^ detection by aequorin may have caused this discrepancy; however, this needs to be verified.

The effects of the loss of Cch1 and Mid1 in *O. parapolymorpha* were very similar: these mutations similarly affected the dynamics of the [Ca^2+^]_cyt_ rise in the response to amiodarone and SDS, improved SDS resistance and slightly decreased survival in the presence of amiodarone. This indicates that these effects are related to the function of the Cch1/Mid1 complex but not to hypothetical functions of Cch1 or Mid1 outside the complex even if such functions exist.

In the strain with the wild-type Ca^2+^ transport system, amiodarone induced a rapid [Ca^2+^]_cyt_ rise, which started within the first two minutes of incubation and continued for more than 10 min until it was probably compensated by the action of ion transporters sequestering Ca^2+^ in the vacuole. The loss of the HACS components caused some delay in the [Ca^2+^]_cyt_ rise that might lead to suggestion that amiodarone induces Ca^2+^ entry via the Cch1/Mid1 channel, while the following [Ca^2+^]_cyt_ increase is mediated by an alternative pathway. However this raises a question: why does this alternative pathway not work at the beginning? A more consistent explanation is that the pathway of Ca^2+^ entry in response to amiodarone can be recruited by Cch1/Mid1 and by amiodarone itself, but this recruitment takes some time, causing the delay observed in the mutants lacking Cch1 and Mid1. Notably, the loss of the HACS components did not improve survival rate on amiodarone-containing medium, indicating that the survival does not depend on the delay in the amiodarone-induced [Ca^2+^]_cyt_ rise.

The survival in the presence of amiodarone may involve Hog1-dependent cell cycle regulation as described for SDS hypersensitivity in the *pmc1-*Δ mutant [18] since both SDS and amiodarone induced a [Ca^2+^]_cyt_ rise accompanied with Hog1 activation. At the same time the primary cellular target of amiodarone can be different from that of SDS since SDS induces the immediate [Ca^2+^]_cyt_ rise, while the [Ca^2+^]_cyt_ rise in response to amiodarone starts after some delay.

In yeast, the homolog of mammalian p38 protein kinase Hog1 is implicated into adaptation to high osmolarity conditions, being activated in response to high osmolarity shock (for a review see [32]). It has also been shown that Hog1 is activated in response to caffeine [33]. One of the consequences of its activation is stabilization of the Wee1 protein kinase, which inhibits the G_2_/M cell-cycle transition [34,35]. Thus, both amiodarone and caffeine can cause Hog1 activation but via different pathways. This is probably why caffeine aggravates the inhibitory effect of amiodarone on cell growth (Appendix A).

The delay in the [Ca^2+^]_cyt_ rise lasted longer in the mutant lacking the Pmc1 Ca^2+^ ATPase than in the strain with the wild-type Ca^2+^ transport system, while the loss of Cch1 or Mid1 additionally extended it. As we observed previously, the basal [Ca^2+^]_cyt_ level is increased in the *pmc1-*Δ mutant [26] that hypothetically may suppress Ca^2+^ uptake systems, thus resulting in the delayed response to amiodarone. After prolonged incubation with amiodarone, the [Ca^2+^]_cyt_ level in the *pmc1-*Δ *cch1-*Δ and *pmc1-*Δ *mid1-*Δ mutants was substantially lower than in the *pmc1-*Δ mutant. This correlated with suppression of the *pmc1-*Δ amiodarone hypersensitivity. This might indicate that HACS is directly involved in the amiodarone-induced Ca^2+^ entry; however, in terms of the “indirect involvement” hypothesis expressed above, this occurs via an alternative pathway, whose activity depends on the HACS components. The less steep [Ca^2+^]_cyt_ rise in the strains lacking Pmc1 Ca^2+^ ATPase can be explained by repression of this alternative pathway due to higher basal [Ca^2+^]_cyt_. This suggestion is consistent with the previous observation that increased external Ca^2+^ concentration reduces the amiodarone-induced [Ca^2+^]_cyt_ burst and rescues cells from amiodarone toxicity in *S. cerevisiae* [12].

Thus, here we have characterized the *O. parapolymorpha MID1* gene and shown that its inactivation leads to the same phenotypes as inactivation of *CCH1* encoding the channel-forming subunit of the Mid1/Cch1 complex. The effects of inactivation of these genes on the amiodarone-induced [Ca^2+^]_cyt_ rise were also very similar, which contradicts the previously published results obtained in *S. cerevisiae* [13]. Our data suggest that the Mid1/Cch1 complex is not the conduit for Ca^2+^ entry in response to amiodarone but is involved in the regulation of this process.

## 4. Materials and Methods

### 4.1. Culture Media and Yeast Transformation

Complex media contained 2% Peptone, 1% yeast extract and 2% glucose (YPD) or 1% sucrose (YP-Suc) as a carbon source. The synthetic medium SC-D (2% glucose, 0.67% Yeast Nitrogen Base with ammonium sulfate) was used for the selection of transformants. The Ca^2+^-deficient medium was prepared as described previously [3]. Yeast cells were transformed using the Li-acetate method [36] with some modifications [37].

### 4.2. Yeast Strains

The *O. parapolymorpha* strains used in this study are listed in Table 2. The process of obtaining these strains except DL5-mid1 bearing the *MID1* deletion allele was described previously [19]. The *MID1* deletion was performed as follows. The DNA fragment with inverted recombination arms was obtained by PCR with OpMID1U1 and OpMID1L1 primers using HindIII-digested and self-ligated *O. parapolymorpha* genomic DNA as a template. This fragment was digested with Bsp1407 and inserted between Bsp1407 and PvuII sites of the pAM576 vector. This vector was obtained by replacement of the wild-type *S. cerevisiae LEU2* selectable marker in the AMIpSL1 vector [38] for the modified one described previously [39]. The obtained plasmid pMK4 was digested with HindIII that resulted in a linear construct possessing vector sequence flanked by the recombination arms according to the scheme described previously [40]. Integration of this construct into the *MID1* locus was confirmed by PCR with primer pairs OpMID1AU1-oriA and OpMID1AL1-SL3. Sequences of PCR primers are presented in Appendix A.

Besides *O. parapolymorpha*, *S. cerevisiae* strain BY4742 and its *cch1-*Δ and *mid1-*Δ derivatives [41] were used in this study.

### 4.3. Monitoring of the Cytosolic Ca^2+^ Concentration

The GEM-GECO genetically encoded Ca^2+^ indicator [27] was used to monitor [Ca^2+^]_cyt_ in individual *O. parapolymorpha* cells using flow cytometry as described previously [19,26].

### 4.4. Growth Inhibition and Cell Survival Assays

To determine minimum inhibitory concentrations of SDS and amiodarone, logarithmic-phase cultures were diluted to an OD_600_ of 0.05 in medium containing serial dilutions of each compound in 96-well plates. The plates were incubated at 37 °C in a shaker incubator, and yeast growth was assessed visually by culture turbidity after 20–24 h incubation for SDS or 20–24 and 44–48 h incubation for amiodarone.

To assess the survival rates on amiodarone-containing medium, logarithmic cultures at OD_600_ 0.8–1.0 were spread onto YPD plates with or without amiodarone. Prior to the plating, the cultures were properly diluted to obtain 500–1000 colonies per plate. Plates were incubated at 37 °C, and yeast colonies were counted after 24 h on the control plates or after 48 h on the amiodarone-containing plates. The survival rate was calculated as a ratio of colony numbers on plates with and without amiodarone with respect to the dilution factor.

### 4.5. Analysis of Hog1 Phosphorylation

The alterations in Hog1 phosphorylation were assayed by immunoblotting using anti Phospho-p38 monoclonal antibody (Cell Signaling Technology, Inc., Danvers, MA, USA, cat. #9215). Total Hog1 content was assayed by immunoblotting using polyclonal rabbit antisera against *Escherichia coli*-expressed *O. polymorpha* Hog1. A more detailed protocol of the analysis of Hog1 phosphorylation was described previously [19].

### 4.6. Statistical Analysis

Statistical analysis was conducted using GraphPad Prism 8.0.1. Data are presented as mean ± SD from at least three independent replicates. Two-way analysis of variance (ANOVA) followed by Tukey’s Honestly Significant Difference (HSD) post hoc test was used for multiple comparisons.

## Figures and Tables

**Figure 1 ijms-26-11386-f001:**
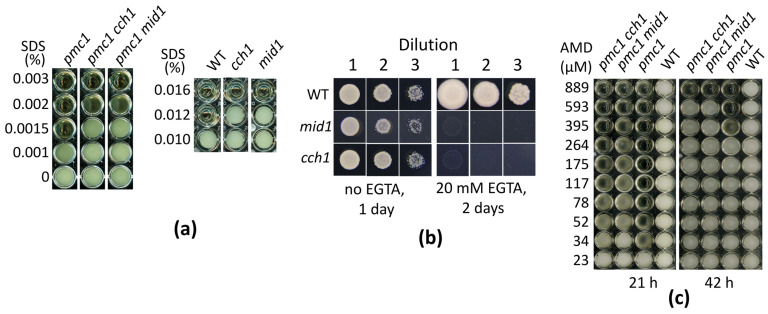
Ability of *O. parapolymorpha* mutants to grow in liquid YPD medium in the presence of SDS (**a**), amiodarone (**c**), or on the solid Ca^2+^-deficient medium (**b**). WT, DL5-LC strain; *cch1*, DL5-cch1 strain; *mid1*, DL5-mid1 strain; *pmc1*, DL5-pmc1-LC strain; *pmc1 cch1*, DL5-pmc1-cch1 strain, *cch1 mid1*, DL5-cch1-mid1 strain. Dilutions 1, 2 and 3: 3 μL of 300-, 3000- and 30,000-fold diluted overnight YPD cultures, respectively, were spotted onto plates with the Ca^2+^-deficient medium with or without EGTA and incubated at 37 °C.

**Figure 2 ijms-26-11386-f002:**
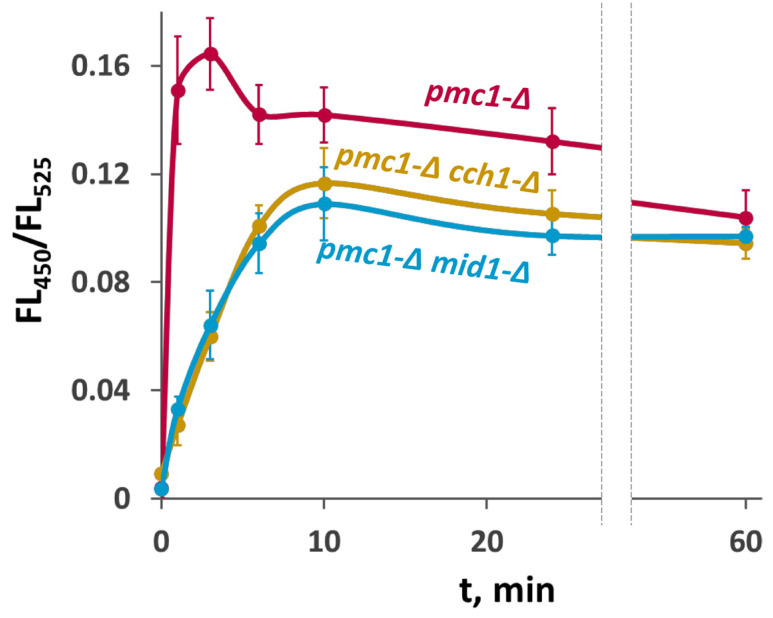
Comparison of effects of inactivation of *CCH1* and *MID1* in the strain lacking *PMC1* on [Ca^2+^]_cyt_ dynamics in response to 0.004% SDS. Changes in [Ca^2+^]_cyt_ were monitored using the GEM-GECO Ca^2+^ indicator by measuring FL_450_/FL_525_ values in individual cells. The median values of 10^4^ cells obtained from 3 replicates were averaged, and standard deviations were calculated.

**Figure 3 ijms-26-11386-f003:**
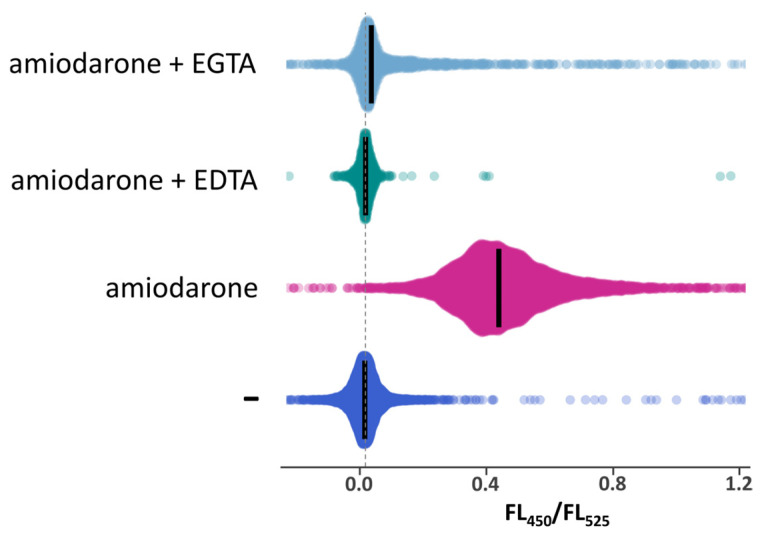
Distribution of cells according to FL_450_/FL_525_ in exponentially grown cultures of the DL5-LC strain, which has the wild-type Ca^2+^ transport system, before (−) or after 10′-incubation with 30 μM amiodarone in the absence or the presence of 10 mM EDTA or 20 mM EGTA. Median values are indicated by bars.

**Figure 4 ijms-26-11386-f004:**
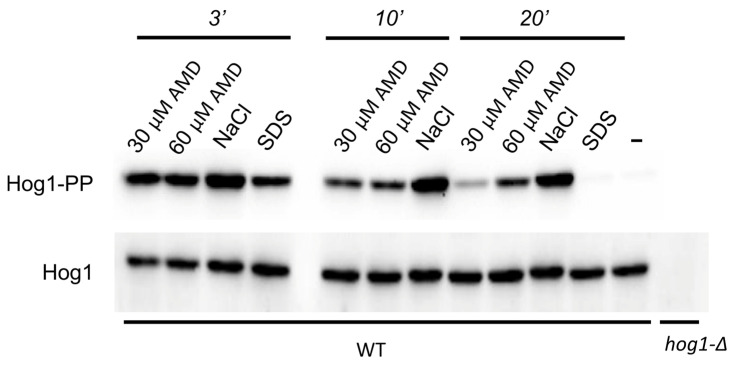
Immunoblotting of phosphorylated (Hog1-PP) and total Hog1 (Hog1) in cell lysates. Exponentially growing culture of the *O. parapolymorpha* DL5-LC strain (WT) was supplemented either with 30 μM amiodarone (30 μM AMD), 60 μM amiodarone (60 μM AMD), 0.6 M NaCl (NaCl), or 0.004% SDS (SDS) and incubated for 3, 10 and 20 min except for samples with SDS, which were incubated for 3 and 20 min. Untreated cells (−) represent the basal level of Hog1 phosphorylation. Cells of the untreated *hog1-*Δ mutant (*hog1-*Δ) were used as a negative control.

**Figure 5 ijms-26-11386-f005:**
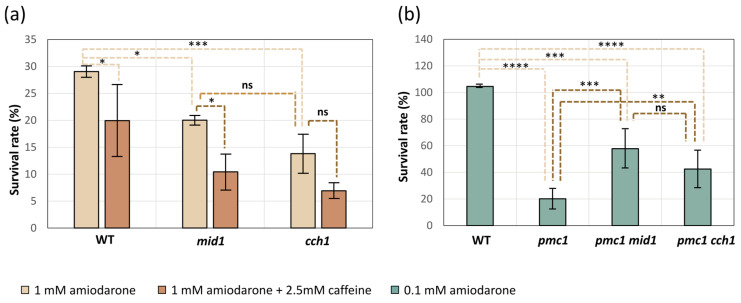
Effects of the *mid1-*Δ and *cch1-*Δ mutations on survival rate on medium containing 1 mM amiodarone or 1 mM amiodarone and 2.5 mM caffeine in strains with wild-type *PMC1* (**a**) and on medium containing 0.1 mM amiodarone in strains with the inactivated *PMC1* (**b**). WT, *cch1*, *mid1*, *pmc1*, *pmc1 mid1* and *pmc1 cch1*: DL5-LC, DL5-mid1, DL5-cch1, DL5-pmc1-LC, DL5-pmc1-mid1 and DL5-pmc1-cch1 strains, respectively. Data represent mean ± SD from at least 3 independent experiments. Significance was determined by two-way ANOVA with Tukey’s multiple comparisons test (* *p* < 0.05, ** *p* < 0.01, *** *p* < 0.001 and **** *p* < 0.0001; ns, not significant).

**Figure 6 ijms-26-11386-f006:**
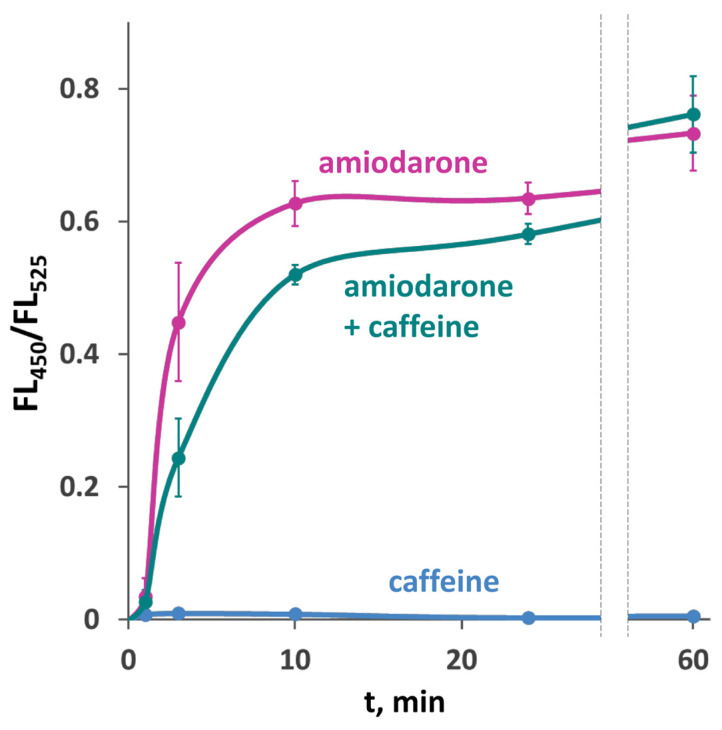
Effect of 10 mM caffeine on [Ca^2+^]_cyt_ rise in response to 30 μM amiodarone in the DL5-LC strain with the wild-type Ca^2+^ transport system. Changes in [Ca^2+^]_cyt_ were monitored using the GEM-GECO Ca^2+^ indicator by measuring FL_450_/FL_525_ values in individual cells. The median values of 10^4^ cells obtained from 3 replicates were averaged, and standard deviations were calculated.

**Figure 7 ijms-26-11386-f007:**
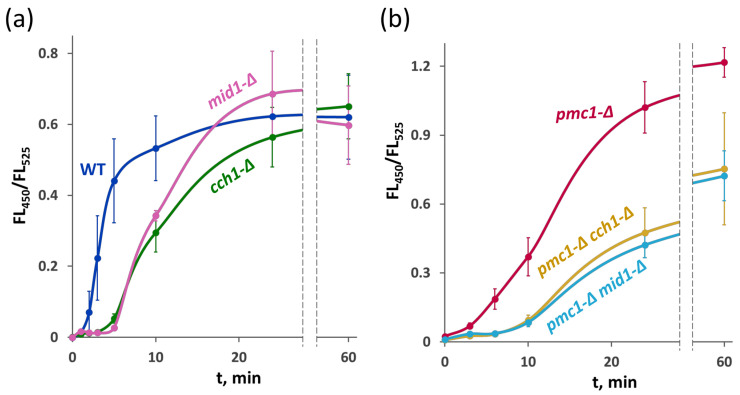
[Ca^2+^]_cyt_ rise in response to 30 μM amiodarone in the strains with the wild-type cytosolic Ca^2+^ sequestration system (**a**) and in the strains lacking the Pmc1 Ca^2+^ ATPase (**b**). Changes in [Ca^2+^]_cyt_ were monitored using the GEM-GECO Ca^2+^ indicator by measuring FL_450_/FL_525_ values in individual cells. The median values of 10^4^ cells obtained from 3 replicates were averaged, and standard deviations were calculated.

**Table 1 ijms-26-11386-t001:** Potential transmembrane regions in Mid1 homologs.

Source	Protein Length	Position of the Hydrophobic Regions	
		N-Terminal	C-Terminal
*O. parapolymorpha*	479	1–13	466–479
*S. cerevisiae*	548	1–14	-
*Y. lipolytica*	640	3–15	627–640
*Sch. pombe*	486	1–16	474–485
*C. neoformans*	623	58–73	-
*C. albicans*	559	1–13	546–559
*A. fumigatus*	642	14–33	621–638

**Table 2 ijms-26-11386-t002:** *O. parapolymorpha* strains used in this study.

Strain	Genotype *	Source
DL5	*leu2 {P_MAL1_-GEM-GECO}*	[26]
DL5-LC	*leu2 {P_MAL1_-GEM-GECO} {LEU2}*	[19]
DL5-cch1	*leu2 cch1::LEU2 {P_MAL1_-GEM-GECO}*	[19]
DL5-pmc1-LC	*leu2 pmc1::loxP {P_MAL1_-GEM-GECO} {LEU2}*	[19]
DL5-pmc1-cch1	*leu2 pmc1::loxP cch1::LEU2 {P_MAL1_-GEM-GECO}*	[19]
DL5-pmc1-mid1	*leu2 pmc1::loxP mid1::LEU2 {P_MAL1_-GEM-GECO}*	This study
DL5-mid1	*leu2 mid1::LEU2 {P_MAL1_-GEM-GECO}*	This study
DL5-hog1	*leu2 hog1::LEU2 {P_MAL1_-GEM-GECO}*	[19]

* Names of genes integrated into unidentified genome loci as part of a plasmid are shown in brackets.

## Data Availability

The original contributions presented in this study are included in the article/Appendix A. Further inquiries can be directed to the corresponding author.

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
