# Peer review of "Inactivation of the High-Affinity Ca2+ Uptake System Delays the Amiodarone-Induced Ca2+ Influx in Yeast Ogataea parapolymorpha"

_ijms, 2025, doi:10.3390/ijms262311386_

Round 1

Reviewer 1 Report

Comments and Suggestions for Authors

The authors investigated the potential role of the two Ca2+ uptake-related genes, MID1 and CCH1, in response to amiodarone treatment. The overall experimental design is good. However, the data presented in the manuscript are too preliminary and do not meet the publication standard. The biggest problem is the lack of quantitative and statistical analyses. As a result, a lot of data are presented as arbitrary, subjective, and descriptive. The second major problem is the lack of proper controls. Specifically,

  1. In the method section (line 312): “…yeast growth was assessed visually by culture turbidity…”. The visual determination is simply not enough. The authors should use a plate reader to quantify the differences. The quantification (including other assays) may also resolve possible differences between Mid1 and Cch1, rather than claiming the effects are the same (lines 108-109).
  2. Other similar wording includes-
  • “equally suppressed” (line 111) – Without quantification, you cannot claim this.
  • “virtually the same” (line 116) – Scientifically speaking, what does this mean? 99% ‘same’? 90% ‘same’?
  • “virtually identical” (line 60).
  • “becomes visibly turbid” (lines 179-180) – Would this not rely on the examiners’ eyes? How do you repeat this experiment?
  1. In the method section, NO statistical analysis was mentioned. This is possibly a major cause of the language used above, instead of saying “no statistically significant difference was observed”, or “the difference is statistically significant (p < xxx)”.
  2. The only place the p-value is mentioned is in the figure legend of Figure 5, but no explanation was given on how the p-value is calculated. In addition, the p-value is indicated in a non-traditional way – using one “*” for a smaller p-value (0.005), and two “**” for a larger p-value (0.05). It implies (to me at least) that the statistical analysis was not done properly.
  3. Figure 1b. It should include strains of pmc1/cch1, and pmc1/mid1. This could further explain the possible role of Mid1 and Cch1.
  4. Figure 1c. It should include strains of cch1 and mid1. It could be used as an additional control, if nothing else.
  5. Figure 2a, where is the WT? It is unclear to me why each figure only shows part of the strains, but not the others.
  6. Figure 3. The x-axis shows the values of 0.0 to 1.2. What does it mean for FL450/FL525 = 0.0? Why are some data points negative?
Comments on the Quality of English Language

Many words used in the manuscript are subjective or arbitrary (such as virtually identical). Please avoid using these types of words in your future manuscripts.

Author Response

We are grateful to the Reviewer for the comments on our manuscript. However we cannot agree with some of the Reviewer's concerns.

Q: 1.    In the method section (line 312): “…yeast growth was assessed visually by culture turbidity…”. The visual determination is simply not enough. The authors should use a plate reader to quantify the differences. The quantification (including other assays) may also resolve possible differences between Mid1 and Cch1, rather than claiming the effects are the same (lines 108-109).

R: We do not agree with the Reviewer's concern in this case. This is a standard method to determine minimal inhibitory concentration and it is not intended to compare growth rates. That is why the results of these experiments are presented in the Figure 1 as a photograph of plate wells, in which yeast cultures were incubated. So, it is easy to assess the presence or absence of cell growth in the wells.

Q: 2.    Other similar wording includes-

    “equally suppressed” (line 111) – Without quantification, you cannot claim this.

R: The sentence has been re-written

Q:     “virtually the same” (line 116) – Scientifically speaking, what does this mean? 99% ‘same’? 90% ‘same’?

R: "virtually the same" has been replaced with "similar"

Q:    “virtually identical” (line 60).

R: Replaced with "very similar"

Q:    “becomes visibly turbid” (lines 179-180) – Would this not rely on the examiners’ eyes? How do you repeat this experiment?

R: This concern probably arose due to misunderstanding, since the mentioned sentence describes not an experiment but observation on solubility of amiodarone in culture medium.

Q: 3.    In the method section, NO statistical analysis was mentioned. This is possibly a major cause of the language used above, instead of saying “no statistically significant difference was observed”, or “the difference is statistically significant (p < xxx)”.

R: Such section has been added to the Materials and Methods section.

Q: 4.    The only place the p-value is mentioned is in the figure legend of Figure 5, but no explanation was given on how the p-value is calculated. In addition, the p-value is indicated in a non-traditional way – using one “*” for a smaller p-value (0.005), and two “**” for a larger p-value (0.05). It implies (to me at least) that the statistical analysis was not done properly.

R: The figure has been edited as the reviewer suggested.

Q: 5.    Figure 1b. It should include strains of pmc1/cch1, and pmc1/mid1. This could further explain the possible role of Mid1 and Cch1.

R: We disagree with this comment. This figure shows the phenotype exerted by inactivation of MID1 and CCH1 in the strain with wild-type PMC1. An increased sensitivity to Ca2+ shortage demonstrates that same as Cch1, Mid1 is responsible for Ca2+ uptake at low Ca2+ concentrations, which means that O. parapolymorpha MID1 is function is the same as that of its S. cerevisiae homolog. Only this conclusion was made based on these results and it does not require assessing the sensitivity to Ca2+ shortage in the strains with pmc1 mutation.

Q: 6.    Figure 1c. It should include strains of cch1 and mid1. It could be used as an additional control, if nothing else.

R: In our experiments, which were not included in the manuscript, the cch1 and mid1 single mutants, as well as the wild-type strain were able to grow at all amiodarone concentrations used for this experiment. This can also be concluded from the data presented in the Figure 5 and in the supplementary materials. So, the result would be the same as in the wild-type strain. We could not use higher amiodarone concentration due to the reason explained in the manuscript. That is why we think that including these mutants in the figure is not reasonable.

Q: 7.    Figure 2a, where is the WT? It is unclear to me why each figure only shows part of the strains, but not the others.

R: Each figure represents an experiment with a specific task, which may not require all the strains to be analyzed. All the strains were studied in the experiments presented on the panels c and d (Figure 7a and b in new version of the manuscript). In case of Figure 2a, the experiment studying cytosolic Ca2+ dynamics in response to SDS in the wild type strain have been described in our previous paper (https://www.mdpi.com/1422-0067/25/21/11450). Here we compare the effects of cch1 and mid1 mutations in the strain with inactivated PMC1.

Q: 8.    Figure 3. The x-axis shows the values of 0.0 to 1.2. What does it mean for FL450/FL525 = 0.0? Why are some data points negative?

R: FL450 and FL525 were calculated by subtraction of estimated autofluorescence in each cell. The calculation method has been briefly described in Materials and Methods and more detailed description can be found in our previous paper (https://www.mdpi.com/1422-0067/23/17/10004). The value 0 means that FL450 value does not differ from the calculated autofluorescence; the negative values mean that it is lower than the calculated autofluorescence.

Reviewer 2 Report

Comments and Suggestions for Authors

Rev Kulakova et al., IJMS 2025

Authors analyze the effect of amiodarone, the antiarrhythmic drug, on the growth and calcium uptake of various calcium transport defective mutants of Ogatea parapolymorpha and compare it with the effects of sodium dodecyl sulphate, the  synthetic surfactant. This repeats some experiments which were performed using Saccharomyces cerevisiae yeast with different method and does not add much more experiments to the better understanding of amiodarone action. Authors should make corrections of several mistakes, correct presentation and explain better in Discussion why this work is important.

General comments

The Introduction must be enriched, what is known about amiodarone action in humans?

Discussion must be enriched, what is a Hog1 pathway, what is caffeine, why is important here, how these results advance our understanding about amiodarone action.

Discussion should start with presenting more general problem which was studied here.

Discussion should end with the paragraph giving conclusions what these studies add to the better understanding of amiodarone action and future perspectives.

Genes were deleted (or disrupted?) not proteins inactivated.

Specific comments

Abstract

P1, L13 it is not completely clear, what is “This”?. It this means deletion of CCH1 or MID1 gene? Than the sentence is contradictory to the next sentence. At this point is also not clear, what is the sequestration system? Only uptake system was presented so far.

Introduction

P1, L37, except what? Something is missing. Which Figure is referred here? There is no Figure 1 in  the Introduction.

P2, L42-48 add more what is known about amiodarone action in human cells.

P2, L49 Ogatea parapolymorpha  , full name should be given here, first time in the main text.

P2, L51 Saccharomyces cerevisiae, full name should be given here, first time in the main text.

P2, L53 signaling and lack of PMC1 prevents Ca 2+ sequestration in the vacuole.  What is calcineurin? It was not introduced so far.

P,  L58 “Here” should start the last paragraph in which reader can find what was the goal of study, what was studied and then brief results.

Results

P3, L65 Deletion of MID1 suppresses SDS hypersensitivity caused by lack of the Pmc1…

P3, L67 O. parapolymorpha yeasts

P3, L77-84 why Candida albicans is not described in the main text? Then could be  C. albicans in the Table 1, as other species.

P3, L81, 82 amino acid residue

P3, L84 contradicts the role as secretory signal.

Table 1

  1. pombe

P3, L93 O. parapolymorpha. Alternatively  give other species which were studied.

P3 L99, grow in aa a medium with chelated Ca 2+ ions. Be more precise, not enigmatic.

Figure 1

EGTA abbreviation should be introduced. Possibly also in the main text.

Figure 2

Part (a) is referred  in chapter 2.2, P4. Part (b) is referred after Figure 3 and 4 in chapter 2.4. Part (c) and (d) is referred after Figure 5 on P8, chapter 2.5. This is confusing, Figures should be referred in the same order as they appear according to the number in the text. Split Figure 1 into two or more figures as needed. Figures are not build according the method used but according to the logic flow in the text.

The title is focused on the method instead of the merit, the concentration of Ca 2+ ions in the cytoplasm. Rather: Dynamics of [Ca 2+]cyt…On the end of the legend add explanation that the [Ca 2+]cyt was monitored as FL450/FL525 ratio of GEM-GECO.

L124 wild-type (WT) DL5-LC strain. If all strains used are isogenic it is sufficient to give this information once in the main text and later give only relevant genotype. Repeating some many times these long names does not help. The reader can find detailed names in Table2.

Add WT into (b) panel graph

P6, L 139 taken up, transported, not absorbed. It is active transport.

P6, L144 And so what? Conclusion is lacking.

Figure 3

P6, L149 wild-type DL5-LC strain…. Amiodarone (AMD) in the absence or presence of…

The abbreviation AMD must be introduced in proper site.  Authors can consider to introduce abbreviation for amiodarone (AMD) in the whole text , as is SDS.

Abbreviations EGTA and EDTA must be explained, introduced in the main text.

Figure 4

P6, L155 wild-type (WT) DL5-LC strain

P6,L 156, Amiodarone (AMD) should be introduced. AMD should be added in the upper part of the figure , above each 30 and 60.

P6, L157, which was incubated

P6, L158 Cells of the untreated hog1- Δ mutant

hog1 is  in incorrect place. Description of strain is now together with stress factor. The line and WT should be added in the bottom of the figure and hog1- Δ  moved to the bottom right corner with small line above.

P6, L161 Deletion of MID1 and CCH1 is sufficient

P7, L168  add reference to Figure S2

P7, L174g rowing cultures of which OD?

P7, L176 rather Figure 5a, see below

P7, L177 colony growth on the medium supplemented with amiodarone

P7, L186 Figure 5b and 1c

P7, L189 what is caffeine doing, why is important here?

P7, L196 Figure 5a and S3. Conclusion is lacking.

Figure 5

Split Figure into two panes a and b. Move the color legend to the left. Change the legend accordingly.

This was probably already proposed by some authors since 5b is in P8, L224.

Discussion

P8, L227, expression of S. cerevisiae MID1… Genes are expressed, proteins are synthesized of produced.

P8, L228 inactivation of gene encoding Mid1

P8, L233 why GEM-CECO is better than aequorin?

P8, L240 maybe caffeine interferes with Ca 2+ detection by GEM-GECO? What caffeine is doing, why is important here? More discussion is needed.

P8, L242 inactivation of genes encoding CCh1 and Mid1

P9, L264 what is “this mutant” pmc1, mid1, cch1?

Materials and Methods

P9, L285, 286 are genes MID1 and CCH1 and HOG1 disrupted or deleted? Be clear here and in the text, not confuse readers.

Table 2

Add the new panel, Source, source of each strain.

P10, L315 what means properly? To the same OD or to the same number of cells in1 ml?

In the Figure S2 control panel with 0 AMD we can see that cell suspensions were not properly diluted, more cells in upper panel with O.p. and less cells in lower panel with S.c.. The same in Figure S3a.

Figure S3.

Title is not correct, since colonies are bigger in panel O.p. AMD than in panel AMD+caffeine.

Add temperature and how many days plates were incubated.

Author Response

We greatly appreciate such thorough analysis of our manuscript. We hope that we could address all the Reviewer's concerns in this version of the manuscript. Point-by-point response to the Reviewer's comments is given below.

Q: The Introduction must be enriched, what is known about amiodarone action in humans?

R: The Introduction has been edited according to the reviewer's recommendation

______________

Q: Discussion must be enriched, what is a Hog1 pathway, what is caffeine, why is important here, how these results advance our understanding about amiodarone action.

Discussion should start with presenting more general problem which was studied here.

Discussion should end with the paragraph giving conclusions what these studies add to the better understanding of amiodarone action and future perspectives.

R: The Discussion has been edited according to the reviewer's recommendation

___________________

Q: Genes were deleted (or disrupted?) not proteins inactivated.

R: The genes were inactivated by replacement of a portion of their ORFs with a selectable marker or with the loxP sequence. Technically, this is both, deletion and disruption. That is why we prefer to use the term "inactivation" in case of genes. In case of proteins, the term "inactivation" has been replaced with "loss" or "lack".   

___________________

Q: P1, L13 it is not completely clear, what is “This”?. It this means deletion of CCH1 or MID1 gene? Than the sentence is contradictory to the next sentence. At this point is also not clear, what is the sequestration system? Only uptake system was presented so far.

R: Actually, "This" refers to all items of the previous sentence: loss of Cch1 or Mid1 resulting in delayed calcium rise. In other words, the loss of Cch1, the loss of Mid1, and the delay "have negligible effect …".

The calcium sequestration system is mentioned in the next sentence. This part of the Abstract has been re-phrased. We hope it is more clear now.  

____________________

Q: P1, L37, except what? Something is missing. Which Figure is referred here? There is no Figure 1 in  the Introduction.

R: We understand that it is confusing, but Fig1 is a protein name.

_____________________

Q: P2, L42-48 add more what is known about amiodarone action in human cells.

R: Relevant information on amiodarone action in humans has been added.

_____________________

Q: P2, L49 Ogatea parapolymorpha  , full name should be given here, first time in the main text.

R: In this case, "Ogataea" refers to O. polymorpha and O. parapolymorpha, since this phenotype was revealed in both species. We added the Ogataea parapolymorpha full name when it is first mentioned.

Q: P2, L51 Saccharomyces cerevisiae, full name should be given here, first time in the main text.

  1. This has been corrected.

_____________________

Q: P2, L53 signaling and lack of PMC1 prevents Ca 2+ sequestration in the vacuole.  What is calcineurin? It was not introduced so far.

R: The explanation and references have been added.

_____________________

Q: P,  L58 “Here” should start the last paragraph in which reader can find what was the goal of study, what was studied and then brief results.

R: This paragraph has been edited.

_____________________

Q: P3, L65 Deletion of MID1 suppresses SDS hypersensitivity caused by lack of the Pmc1…

R: This has been corrected.

_____________________

Q: P3, L67 O. parapolymorpha yeasts

R: The name Ogataea refers to both O. polymorpha and O. parapolymorpha

_____________________

Q: P3, L77-84 why Candida albicans is not described in the main text? Then could be  C. albicans in the Table 1, as other species.

R: The names of "other species" were added to the text.

_____________________

Q: P3, L81, 82 amino acid residue

P3, L84 contradicts the role as secretory signal.

R: Both items have been corrected

_____________________

Q: Table 1

    pombe

R: We were not sure what was wrong. Possibly, the problem was solved when we added the species names to the text.

_____________________

Q: P3, L93 O. parapolymorpha. Alternatively  give other species which were studied.

R: The species names have been added.

_____________________

Q: P3 L99, grow in aa a medium with chelated Ca 2+ ions. Be more precise, not enigmatic.

Figure 1 - EGTA abbreviation should be introduced. Possibly also in the main text.

R: This has been corrected.

_____________________

Q: Figure 2   Part (a) is referred  in chapter 2.2, P4. Part (b) is referred after Figure 3 and 4 in chapter 2.4. Part (c) and (d) is referred after Figure 5 on P8, chapter 2.5. This is confusing, Figures should be referred in the same order as they appear according to the number in the text. Split Figure 1 into two or more figures as needed. Figures are not build according the method used but according to the logic flow in the text.

The title is focused on the method instead of the merit, the concentration of Ca 2+ ions in the cytoplasm. Rather: Dynamics of [Ca 2+]cyt…On the end of the legend add explanation that the [Ca 2+]cyt was monitored as FL450/FL525 ratio of GEM-GECO.

L124 wild-type (WT) DL5-LC strain. If all strains used are isogenic it is sufficient to give this information once in the main text and later give only relevant genotype. Repeating some many times these long names does not help. The reader can find detailed names in Table2.

Add WT into (b) panel graph

R: The figure has been edited according to the Reviewer's request. The panel (b) is a separate figure now. The strain used in the experiment is mentioned in the figure legend.

_____________________

Q: P6, L 139 taken up, transported, not absorbed. It is active transport.

R: This has been corrected

_____________________

Q: P6, L144 And so what? Conclusion is lacking.

R: Conclusion has been added.

_____________________

Q: Figure 3

P6, L149 wild-type DL5-LC strain…. Amiodarone (AMD) in the absence or presence of…

The abbreviation AMD must be introduced in proper site.  Authors can consider to introduce abbreviation for amiodarone (AMD) in the whole text , as is SDS.

Abbreviations EGTA and EDTA must be explained, introduced in the main text.

R: These items have been corrected. Formally, DL5-LC is not a wild-type strain. We added " which has the wild-type Ca2+ transport system" instead.

_____________________

Q: Figure 4

P6, L155 wild-type (WT) DL5-LC strain

P6,L 156, Amiodarone (AMD) should be introduced. AMD should be added in the upper part of the figure , above each 30 and 60.

P6, L157, which was incubated

P6, L158 Cells of the untreated hog1- Δ mutant

hog1 is  in incorrect place. Description of strain is now together with stress factor. The line and WT should be added in the bottom of the figure and hog1- Δ  moved to the bottom right corner with small line above.

R: The figure and its legend have been changed according the reviewer's requests.

_____________________

Q: P6, L161 Deletion of MID1 and CCH1 is sufficient

R: This has been corrected.

_____________________

Q: P7, L168  add reference to Figure S2

R: The reference has been added.

_____________________

Q: P7, L174 growing cultures of which OD?

R: This information has been added.

_____________________

Q: P7, L176 rather Figure 5a, see below

R: This has been corrected.

_____________________

Q: P7, L177 colony growth on the medium supplemented with amiodarone

R: This has been corrected.

_____________________

Q: P7, L186 Figure 5b and 1c

R: This has been corrected.

_____________________

Q: P7, L189 what is caffeine doing, why is important here?

R: This paragraph was detached into a separate subsection.

_____________________

Q: P7, L196 Figure 5a and S3. Conclusion is lacking.

R: A conclusion has been added.

_____________________

Q: Figure 5

Split Figure into two panes a and b. Move the color legend to the left. Change the legend accordingly.

This was probably already proposed by some authors since 5b is in P8, L224.

R: The figure was split into two panels.

_____________________

Q: P8, L227, expression of S. cerevisiae MID1… Genes are expressed, proteins are synthesized of produced.

P8, L228 inactivation of gene encoding Mid1

R: Both items has been corrected

_____________________

Q: P8, L233 why GEM-CECO is better than aequorin?

R: The explanation has been introduced.

_____________________

Q: P8, L240 maybe caffeine interferes with Ca 2+ detection by GEM-GECO? What caffeine is doing, why is important here? More discussion is needed.

R: We did not see much difference in GEM-GECO fluorescence when cells were incubated with amiodarone in presence of caffeine. Otherwise, our results would confirm the results obtained with aequorin. Actually, we don't know whether caffeine inhibits aequorin or not. This is just our suggestion, which could explain the discrepancy.

A sentence mentioning mechanism of action of caffeine in yeast was introduced into this part of the Discussion.

_____________________

Q: P8, L242 inactivation of genes encoding CCh1 and Mid1

R: This has been corrected.

_____________________

Q: P9, L264 what is “this mutant” pmc1, mid1, cch1?

R: This has been specified.

_____________________

Q: P9, L285, 286 are genes MID1 and CCH1 and HOG1 disrupted or deleted? Be clear here and in the text, not confuse readers.

R: As it was mentioned above, technically, both terms, deletion and disruption, can be used in case of these mutations. "Disruption" has been replaced with "deletion" in this sentence.

_____________________

Q: Add the new panel, Source, source of each strain.

R: Panel Source has been added

_____________________

Q: P10, L315 what means properly? To the same OD or to the same number of cells in1 ml?

R: The dilution was adjusted in each case to obtain 500-1000 colonies per plate. In preliminary experiments, we found this number of colonies per plate is countable and allows obtaining reproducible results. We applied increased amount of cell suspension when we expected lower survival rated. The description of this procedure has been re-written. We hope it is more coherent now.  

_____________________

Q: In the Figure S2 control panel with 0 AMD we can see that cell suspensions were not properly diluted, more cells in upper panel with O.p. and less cells in lower panel with S.c.. The same in Figure S3a.

R: We would like to note that the objective of this experiment was not to compare the numbers of colonies in O.p. and S.c. This figure shows that at low concentrations, amiodarone only inhibits growth rate, while at higher concentrations it may cause death of some portion of cells in population. Only these facts were discussed in the manuscript.

_____________________

Q: Figure S3. Title is not correct, since colonies are bigger in panel O.p. AMD than in panel AMD+caffeine.

Add temperature and how many days plates were incubated.

R: Following the Reviewer's request expressed in a comment above, we split this figure. The new figures have different legends. The incubation temperatures are indicated now.

Round 2

Reviewer 1 Report

Comments and Suggestions for Authors

I am satisfied with the revision. The manuscript can now be accepted for publication.

Reviewer 2 Report

Comments and Suggestions for Authors

The manuscript was greatly improved and now is suitable for publication.

It would be less confusing if P1, L38  reads: except Fig1 protein [7],